# Mitochondrial Distress in Methylmalonic Acidemia: Novel Pathogenic Insights and Therapeutic Perspectives

**DOI:** 10.3390/cells11193179

**Published:** 2022-10-10

**Authors:** Svenja Aline Keller, Alessandro Luciani

**Affiliations:** Mechanisms of Inherited Kidney Diseases Group, Institute of Physiology, University of Zurich, 8057 Zurich, Switzerland

**Keywords:** epithelial cell distress, metabolism, mitochondria, mitophagy, oxidative stress, kidney tubule, signaling

## Abstract

Mitochondria are highly dynamic, double-membrane-enclosed organelles that sustain cellular metabolism and, hence, cellular, and organismal homeostasis. Dysregulation of the mitochondrial network might, therefore, confer a potentially devastating vulnerability to high-energy-requiring cell types, contributing to a broad variety of hereditary and acquired diseases, which include inborn errors of metabolism, cancer, neurodegeneration, and aging-associated adversities. In this Review, we highlight the biological functions of mitochondria-localized enzymes, from the perspective of understanding the pathophysiology of the inherited disorders destroying mitochondrial homeostasis and cellular metabolism. Using methylmalonic acidemia (MMA) as a paradigm of mitochondrial dysfunction, we discuss how mitochondrial-directed signaling pathways sustain the physiological homeostasis of specialized cell types and how these may be disturbed in disease conditions. This Review also provides a critical analysis of molecular underpinnings, through which defects in the autophagy-mediated quality control and surveillance systems contribute to cellular dysfunction, and indicates potential therapeutic strategies for affected tissues. These insights might, ultimately, advance the discovery and development of new therapeutics, not only for methylmalonic acidemia but also for other currently intractable mitochondrial diseases, thus transforming our ability to modulate health and homeostasis.

## 1. Introduction

Although commonly described as double-membrane organelles devoted to empowering cellular energy, mitochondria play a key role in energy metabolism and, thus, control physiological and organismal homeostasis [1]. These organelles also regulate intracellular calcium and redox balance [2,3] and seem to act as cardinal platforms, toggling the balance between survival and cell death pathways in response to changes in microenvironmental cues, ultimately controlling a wide variety of cellular functions and dictating cell fate decisions [4]. Exciting new discoveries also indicate that mitochondria exchange content and information with other intracellular organelles, through the establishment of membrane contact sites [5]. Consequently, mutations in nuclear and mitochondrial genes that affect mitochondrial homeostasis and functions might pose a potentially devastating threat to many different cell types, fueling metabolic diseases, cancer, neurodegeneration, and aging-related pathologies [6]. Therefore, maintenance of the mitochondrial network offers promising targetable approaches for the development of transformative therapeutics for rare and highly prevalent diseases [7].

Over the last two decades, studies of rare inherited genetic diseases, in combination with advances in technology and high-throughput omics, have meaningfully contributed to deciphering the fundamental principles governing the homeostasis of mitochondria and, hence, their biological functions in the context of physiology and disease. These pathway paradigms, revealed from the genetics of mitochondrial diseases, have substantially helped to understand the pathogenesis of other common human disorders in which mitochondrial dysfunction has been involved as a secondary event in disease development and progression. In agreement, genome-wide association studies (GWAS) have recently substantiated the contribution of variants in mitochondrial genes to disease risk within the population.

Epithelial cells lining the proximal tubule (PT) of the kidney are enriched in mitochondria, to ensure sufficient energy production for their transport function and overall kidney homeostasis. Disruption of mitochondrial homeostasis drives tubular dysfunction, leading to various degrees of kidney disease [8]. Methylmalonic acidemia (MMA), a rare inborn error of metabolism caused by deficiency of the enzyme methylmalonyl-CoA mutase (MMUT), severely affects cell types/organs heavily reliant on energy demand, e.g., the brain, and eyes, kidney, liver, heart, and endocrine and skeletal systems [9]. Patients usually display hypotonia, anorexia, vomiting, respiratory distress, failure to thrive, developmental and psychomotor delay, liver dysfunction, and kidney manifestations [9].

Taking the dysfunction of the mitochondrial-matrix-residing metabolic enzyme methylmalonyl-coenzyme A mutase (MMUT) as a paradigm of a mitochondrial disease causing kidney tubular damage, the purpose of this Review is to highlight the role of mitochondria, not only as an energy powerhouse but also as a hub for the signaling and homeostasis pathways. This review also places a special emphasis on the emerging advances linking loss-of-function mutations in *MMUT* to defects in the autophagy and mitochondrial quality control systems, leading to mitochondrial distress and cellular damage, resulting in multiorgan dysfunctions, and, ultimately, causing life-threatening manifestations. In the concluding section, we examine how the exploitation of the regulatory pathways targeting cellular adversities linked to MMUT dysfunction may lend itself to new therapeutic paradigms.

## 2. Mitochondria: From Structure to Cellular Physiology

Mitochondria are dynamic and plastic organelles bearing multiple copies of their own bacterium-derived mitochondrial DNA (mtDNA), as circular double-stranded DNA molecules. The mitochondrial proteome is composed of approximately 1500 proteins [10,11], which are encoded by nuclear DNA, translated in the cytosol, and transported through the mitochondrion by dedicated import, processing, and assembly systems. Conversely, mtDNA codes roughly 1% of the mitochondrial proteins and enzymes that maintain the functions of the mitochondrial oxidative phosphorylation system (OXPHOS) [12].

Two phospholipid bilayers, i.e., the outer and inner membranes, delimit two distinct mitochondrial sub-compartments: the inner membrane and the mitochondrial space. Beyond its role in controlling the diffusion of ions and small proteins, the mitochondrial outer membrane might act as a signaling hub for metabolic transduction cascades or be involved in exchanging metabolites with other intracellular organelles through the formation of contact sites. The mitochondrial inner membrane contains specific invaginations, called cristae, which regulate the oxidative phosphorylation and, thus, maintain the production of energy [12,13,14].

Beyond their role in energy production and homeostasis, mitochondria play a major role in the regulation of fatty acid, amino acid, and nucleotide metabolism [15], iron-sulfur cluster and cofactor biogenesis [16], and assembly of precursor proteins that are synthetized on cytosolic ribosomes [17]. Furthermore, mitochondria can serve as signaling platforms that regulate programmed cell death and innate immunity [6].

Mitochondria also maintain the storage of versatile signaling metabolites. For example, through the formation of membrane contact sites with the endoplasmic reticulum (ER), mitochondria preserve the homeostasis of calcium in response to different stimuli and stresses and, hence, the control of several important processes within the cell [18].

## 3. Mechanisms of Mitochondrial Quality Control

Mammalian cells (attempt to) cope with the perturbations of homeostasis through dedicated quality control and surveillance systems. Through fission and fusion events, mitochondria can rapidly adjust their morphology and, thereby, ensure the requisite number of functional organelles in response to different stresses [19]. A diverse array of mitochondrial fusion proteins, i.e., the dynamin-like mitofusin 1 (MFN1), MFN2, and optic atrophy factor 1 (OPA), can eventually counteract mitochondrial stress through the fusion of damaged and/or dysfunctional mitochondria with healthy mitochondria [20]. On the other hand, the proteins involved in mitochondrial fission, such as the cytosolic dynamin-like protein DRP1 and its outer mitochondrial membrane partners, remove exhausted parts of the mitochondrial network, which are then delivered to lysosome-directed degradation pathways [21]. This crosstalk between membrane dynamics and mitochondrial quality control has been extensively reviewed elsewhere [22,23]. Here, in this Review, we focus on the “eat-me” signaling cascades regulating the removal of damaged and/or dysfunctional mitochondria by autophagy–lysosome degradative pathways.

Metazoans, such as the worm *Caenorhabditis elegans* and fruit fly *Drosophila melanogaster*, and mammals can remove exhausted and/or potentially harmful mitochondria, through the selective activation of an evolutionary-conserved and self-regulated process, aptly coined mitochondrial autophagy/mitophagy. This surveillance system results in the engulfment and, hence, sequestration of damaged and/or dysfunctional organelles within double-membrane organelles, called autophagosomes, which fuse with lysosomes for cargo degradation and recycling [24,25], ultimately safeguarding homeostasis and physiology at the cellular, physiological, and organismal levels.

The priming of damaged and/or dysfunctional mitochondria for autophagic degradation requires the engagement of two gene-encoding proteins that are mutated and functionally defective in early-onset recessive Parkinson’s disease—the phosphatase and tensin homologue (PTEN)-induced putative kinase 1 (*PINK1*), which encodes for a mitochondrially localized kinase, and *PARK2*, with a protein product, Parkin, that is a cytosolic E3 ubiquitin ligase [20,26]. In healthy mitochondria, the PINK1 protein is transported via the TOM and TIM23 complexes into the mitochondrial inner membrane, processed, and subsequently cleaved by the matrix processing peptidase (MPP) and presenilin-associated rhomboid-like protein (PARL) [20,27,28]; it is then, hence, retro-translocated into the cytosol and degraded by the ubiquitin-proteasome system [29]. Conversely, once mitochondria are functionally damaged, the loss of mitochondrial-membrane potential promotes the stabilization of PINK1 at the outer mitochondrial membrane and, hence, the exposure of its catalytic domain towards cytosol. This enhances the phosphorylation and, consequently, the activation of the E3 ubiquitin ligase Parkin, which triggers the ubiquitination of other OMM proteins, culminating in the recruitment of autophagy-initiating factors (e.g., optineurin, NDP52, and the engulfment of damaged and/or dysfunctional mitochondria within autophagy–lysosome degradation pathways). This “eat me” signaling cascade is tonically suppressed by a deubiquitylase USP30, through the removal of Parkin-mediated ubiquitylation, thereby modulating the PINK-substrate availability and mitophagy initiation [30]. Beyond its role in mitophagy and mitochondrial clearance, the recruitment and activation of Parkin by PINK1 might also promote the proteosome-induced degradation of PARIS—a transcriptional repressor of PGC1—ultimately stimulating mitochondrial biogenesis and function [31,32]. By integrating biogenesis programs with degradation pathways, PINK1–Parkin-directed mitophagy might, thus, safeguard the maintenance of a healthy mitochondrial reticulum, ultimately preserving its biological functions in energy metabolism and homeostasis [32].

Beyond ubiquitin-driven degradation, other OMM-associated mitophagy receptors, such as BNIP3 (BCL2/adenovirus E1B 19 kDa protein‒interacting protein 3), NIX (NIP3‒like protein X)/BNIP3L, and FUNDC1 (FUN14 domain‒containing 1), promote the binding of mitochondria with MAP1LC3B and GABA_A_-receptor-associated protein (GABARAP), through atypical or typical MAP1LC3B-interacting motifs, and, thus, deliver the damaged and/or dysfunctional organelles to the autophagy–lysosome degradative systems [33,34,35,36]. In a similar vein, the delocalization of the phospholipid cardiolipin away from the IMM and its redistribution to OMM may initiate the signaling cascade that triggers the engulfment of damaged and/or dysfunctional mitochondria by LC3-flagged autophagosomes, their degradation by lysosomes, and, hence, the maintenance of homeostasis in response to mitochondrial damage [37].

Though mitophagy seems to be the dominant mechanism for mitochondrial quality control, a recently emerging piecemeal mitophagy mechanism of mitochondrial quality control has been described. The latter involves the release of small vesicles excised from the mitochondria, called mitochondrial-derived vesicles (MDV) [38], which subsequently fuse with the lysosome for the final disposal and recycling. Thus, MDVs might serve as a first round of defense for mitochondria to remove damaged proteins, preventing the complete collapse of the organelle. Additionally, damaged and/dysfunctional mitochondria were found to be secreted in the extracellular milieu of different cell types, in response to mitochondrial and/or oxidative stress, as well as in disease conditions such as asthma and neurodegeneration diseases [39]. This clearance pathway, aptly called the autophagic secretion of mitochondria, requires different upstream ATG proteins involved in autophagosome biogenesis, while being independent of the mATG8-conjugation system and lysosome degradation [39]. Taken together, these studies suggest that a diverse string of quality control mechanisms might ensure the maintenance of a functional pool of mitochondria and, hence, their biological functions in energy metabolism, cellular physiology, and organismal homeostasis.

## 4. Pathophysiology of Tubular Damage in MMA: The Role of Defective Mitophagy

The maintenance of a healthy and functional mitochondrial network is particularly crucial for terminally differentiated cells that highly rely on aerobic metabolism for sustaining their functions. Imbalances in mitochondrial activities can lead to metabolic dysfunction disease.

Organic acidemias constitute a large group of inherited, life-limiting disorders of intermediary metabolism, mostly due to defects in the enzymes regulating amino acid catabolism. Methylmalonic acidemia (MMA; MIM #251000)—the most common form of organic acidemias—is caused by inactivating mutations in the *MMUT* gene encoding methylmalonyl-CoA mutase (MMUT), which is a (vitamin B_12_-dependent) mitochondrial enzyme that catabolizes branched amino acids and certain lipids to succinyl-CoA, ultimately feeding the tricarboxylic acid (TCA) cycle and energy production. Complete or partial deficiency of the MMUT enzyme (the *MMUT*^0^ and *MMUT*^-^ subtypes, respectively) triggers the accumulation of toxic organic acids (e.g., methylmalonic acid, propionic acid, and 2-methylcitric acid) and leads to abnormalities in the mitochondrial network. Augmented levels of methylmalonic acid have recently been described as inhibiting the electron transport chain (ECT) and, thus, impairing the energy metabolism of mitochondria. The disruption of ETC skews the production of ATP and leads to reactive oxygen species (ROS) and mitochondrial distress [40]. Apparently, MMUT deficiency, mitochondrial dysfunction, and oxidative stress can impact each other in feed-forward loops. These mitochondrial alterations can drive life-threatening organ dysfunctions, primarily affecting the brain, eye, liver, and kidney (Figure 1). Clinical manifestations and long-term complications have been expertly reviewed elsewhere [41,42,43]. How, mechanistically, the loss of MMUT enzymatic activity begets mitochondrial distress and tissue damage remains poorly understood.

Studies using kidney tubular cells derived from the urine of MMA patients (hereafter, referred to as MMA cells) suggested that the loss of MMUT enzyme activity leads to an accumulation of damaged and/or dysfunctional mitochondria, which abnormally generate excess reactive oxygen species (ROS). This triggers an exaggerated production of lipocalin 2 (Lcn2)—a small iron-transporting protein largely produced by kidney tubular cells, following cellular damage, that is associated with kidney disease progression [44] and other metabolic disorders [45]. These phenotypes are in line with the observed correlation between mitochondrial dysfunction, ROS overproduction, and augmented levels of LCN2 in a cohort of MMA patients [46,47], further corroborating the pathogenic implication of mitochondria distress in MMA disease. Additionally, such metabolic and mitochondrial abnormalities have also been described in mouse kidney tubular cells, following conditional inactivation of *Mmut*, and/or in the kidney of mice carrying a mutant *Mmut* allele (p.Met698Lys, corresponding to the patient mutation p.Met700Lys) and a knockout Mmut allele (hereafter *Mmut*^KI/KO^), as well as in the liver and kidney of *mmut* (CRISPR/Cas9-mediated)-deficient zebrafish, demonstrating the evolutionary conservation of this connection [41,48].

Despite the identification of mitochondrial and cellular defects associated with MMA in different model organisms and cellular systems, little is known about the molecular underpinnings linking the lack of the MMUT enzyme and the resulting storage of toxic metabolites to mitochondrial dysfunction and kidney tubular damage. Given the accumulation of damaged and/or dysfunctional mitochondria in MMA cells, the absence of the MMUT enzyme might sabotage the PINK1–PRKN-mediated priming of MMA-diseased mitochondria and, hence, their clearance by the autophagy–lysosome degradative pathways [41,48]. Cellular studies suggested that MMA cells display decreases in the number of PRKN^+^ clusters and translocation of PRKN to dysfunctional mitochondria, under both normal and stress-induced conditions [41,48], tamping down the clearance of MMA-diseased mitochondria and triggering a level of mitochondrial dysfunction that triggers cellular distress and tubular damage (Figure 2).

The quality control of dysfunctional organelles, including mitochondria, is regulated by autophagy, an evolutionary-conserved process that catabolizes cellular constitutes with the help of lysosomes. Accumulation of dysfunctional mitochondria in MMA-affected kidney cells suggests that this cellular clearance mechanism is impaired [48]. Dysregulated autophagy–lysosome degradation pathways have been observed in the livers of a muscle transgenic MMA mouse model and patients with MMA [47], suggesting the key role of the mitochondrial quality control and surveillance systems in the context of tissue homeostasis and disease. Further substantiating the link between defective mitophagy and mitochondrial dysfunction, the gain-of-function interventions activating PINK1–PRKN-directed mitophagy enhance the clearance of MMA-damaged mitochondria, ultimately ameliorating their functions and averting cellular distress. Observations in MMA and HAP-1 cells lacking either *PINK1* or *PRKN2* (encoding PRKN) suggest that anomalies in PINK1–PRKN-mediated quality control might intersect the metabolic/mitochondrial abnormalities wrought by MMUT loss, presumably contributing to the pathogenesis of MMA. This idea is supported by comparative studies of PT cells lacking Cox10—a well-established cellular model that faithfully recapitulates a primary mitochondrial respiratory chain disease [49]. The deletion of *Cox10* in PT cells triggered mitochondrial alterations that are not functionally connected to defects in the PINK1–PRKN-mediated clearance and quality control systems, in contrast with the dysregulation triggered by MMUT deficiency.

The quality control of dysfunctional organelles, including mitochondria, is regulated by autophagy, an evolutionary-conserved process that catabolizes cellular constitutes with the help of lysosomes. Accumulation of dysfunctional mitochondria in MMA-affected kidney cells suggests that this cellular clearance mechanism is impaired [48]. Dysregulated autophagy–lysosome degradation pathways have been observed in the livers of a muscle transgenic MMA mouse model and patients with MMA [47], suggesting the key role of mitochondrial quality control and surveillance systems in the context of tissue homeostasis and disease. Further substantiating the link between defective mitophagy and mitochondrial dysfunction, the gain-of-function interventions activating PINK1–PRKN-directed mitophagy enhance the clearance of MMA-damaged mitochondria, ultimately ameliorating their functions and averting cellular distress. Observations in MMA and HAP-1 cells lacking either *PINK1* or *PRKN2* (encoding PRKN) suggest that anomalies in PINK1–PRKN-mediated quality control might intersect the metabolic/mitochondrial abnormalities wrought by MMUT loss, presumably contributing to the pathogenesis of MMA. This idea is supported by comparative studies in PT cells lacking Cox10—a well-established cellular model that faithfully recapitulates a primary mitochondrial respiratory chain disease [49]. The deletion of *Cox10* in PT cells triggered mitochondrial alterations that are not functionally connected to defects in the PINK1–PRKN-mediated clearance and quality control systems, in contrast with the dysregulation triggered by MMUT deficiency. Thus, the interplay between MMA-driven mitochondrial dysfunction and the defective PINK1–PRKN-mediated quality control might detrimentally have a cumulative effect on the integrity of kidney tubular epithelial cells. Further work on the underlying mechanisms of the defective PINK1–PRKN mitophagy signaling pathway is required. Post-translational modifications that might potentially tamp down PINK1 kinase activity, such as S-nitrosylation [50] or methylmalonylation [51], provide new insights into the regulatory steps at play during mitochondrial quality control in specialized epithelial cells.

## 5. Mitochondria and Cellular Quality Control as Therapeutic Targets for MMA

There are no curative treatments for MMA, and the current supportive care approaches have substantially decreased mortality and overall morbidity. A low-protein diet, supplementation with carnitine and Vitamin B12, and transplantation of the liver and/or kidney (as long-term measures), can indeed attenuate the metabolic dysregulation/instability associated with MMA and, hence, maintain an adequate systemic homeostasis. However, the above treatment implementation cannot prevent long-term irreversible (cerebral and systemic) complications [48]. Therefore, there is an unmet need to design and develop novel therapeutic strategies in the early course of this devastating disorder.

Dissecting the regulatory circuits that drive the complex cellular cascade could eventually lead to discovery and development of therapeutic interventions for treating dysregulated homeostasis in MMA. For example, the bioavailable form of coenzyme Q_10_, i.e., ubiquinone, and vitamin E blunted kidney dysfunction in a transgenic mouse model of MMA by targeting mitochondrial dysfunction and redox homeostasis [46]. In agreement, studies combining machine learning tools and a cross-species screening and validation workflow demonstrated that treatment with mitochondria-targeted ROS scavengers MitoTEMPO and MitoQ repairs mitochondrial homeostasis and function in patient cells and alleviates disease-relevant phenotypes in a zebrafish model of MMA [48]. Importantly, both MitoTEMPO and MitoQ did not modify the levels of MMA metabolite in either MMA cells or *mmut*-deficient zebrafish, indicating that pharmacological strategies restore cellular homeostasis downstream of MMUT loss/accumulation of toxic metabolites. Whether these mitochondria-targeted antioxidants might mediate their protective effects in MMA cells through the modulation of mitophagy and cellular quality control systems are under development [48].

Given that loss of MMUT enzyme activity results in defective mitophagy-mediated degradation of damaged and/or dysfunctional mitochondria in MMA-affected cells, approaches aimed at boosting PINK1–Parkin-directed mitophagy and cellular quality control might counteract cellular and metabolic abnormalities driven by the loss of MMUT enzyme in MMA patients. In this case, the use of either synthetic (e.g., enhancer of either PINK1 or Parkin activity, and inhibitor of the ubiquitin-specific protease 30 that negatively modulate the PINK1–Parkin signaling cascade [23] or natural compounds activating directly or indirectly mitophagy (e.g., NAD^+^ precursors, such as nicotinamide riboside (NR), nicotinamide mononucleotide (NMN) and nicotinamide (NAM) (reviewed in [52])); autophagy and mitophagy activators such as spermidine [53], resveratrol [54], and urolithin A [55]; and mitochondrial-stress-response inducers such as actinonin [56] and doxycycline [57] would, thus, represent an attractive strategy for therapeutically treating dysregulated homeostasis in MMA. Indeed, in a new study, Xie and colleagues [58] have used an artificial intelligence (AI)-powered model to screen a library of 3274 naturally occurring compounds and cross-species approaches for the identification of possible mitophagy inducers. Among the AI-selected molecules, two bioavailable compounds, e.g., Kaempferol and Rhapontigenin, seem to stimulate mitophagy and, hence, mitochondria degradation, leading to increased survival and functionality glutamatergic and cholinergic neurons, abrogating amyloid-β and tau pathologies, and improving cognitive functions in nematode and rodent models of Alzheimer’s disease (AD) [58]. Although hyperactivation of mitophagy may occur without unfavorable consequences for health, these studies support the potential value of examining mitophagy inducers as a new drugging pathway for MMA in a clinical setting.

## 6. Concluding Remarks

The maintenance of a healthy mitochondrial network is particularly crucial for cellular, tissue and organismal homeostasis. Imbalances in mitochondrial activities can invariably confer a potentially devastating vulnerability to many different cell types, ultimately causing a broad spectrum of diseases. Inherited defects in mitochondrial-localized proteins and/or enzymes, as exemplified here by MMA, might compromise PINK1–Parkin-mediated mitophagy and mitochondrial quality control, eventually promoting cellular dysfunction that drives tissue damage. Interestingly, and unlike cardiomyocytes of rodent models lacking *Pink1* [59] or *Parkin* [60], which show accumulation of morphologically abnormal mitochondria and heart pathology, patients with MMA rarely display cardiomyopathy [61]. This context and cell type specificity can be most likely due to the compensatory (stress-evoked) mechanisms that coordinate mitochondrial quality control in non-affected tissues or to the cumulative effect of the metabolic perturbations resulting from the absence of MMUT enzyme and defective mitophagy, to reach a high level of mitochondrial dysfunction, which ultimately triggers the onset and progression of disease relevant-phenotypes in MMA.

The molecular mechanisms, by which MMUT deficiency suppresses PINK1–Parkin-mediated mitophagy, remain equally elusive. MMUT deficiency may alter the stability of PINK1 by disabling the interaction with yet-unknown factors that protect PINK1 from processing and degradation. Alternatively, MMUT deficiency may induce stress-related posttranslational modifications, such as S-nitrosylation [50] or aberrant methylmalonylation [51], which inhibits PINK1 kinase activity and, hence, mitophagy-directed degradation pathways.

The use of informative organism models and mitochondria-based functional assays, coupled with improved knowledge of cell biology–disease signatures and the recent advances in high throughput profiling technologies, may ultimately catalyze the development of therapeutics that could eventually treat MMA and other diseases driven by mitochondrial dysfunction.

## Figures and Tables

**Figure 1 cells-11-03179-f001:**
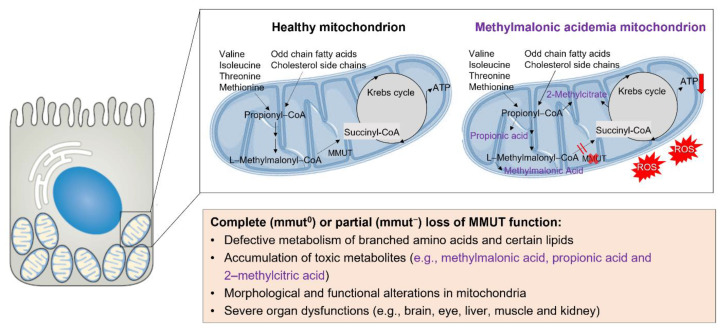
The absence of the enzyme MMUT leads to accumulation of organic acids and mitochondrial abnormalities. Mutations in the *MMUT* gene encoding the mitochondrial enzyme methylmalonyl-coenzyme A mutase, which mediates the terminal step of branched chain amino acid and odd-chain lipid catabolism, trigger the accumulation of metabolites (e.g., methylmalonic acid, propionic acid, and 2-methylcitric acid) and lack of anaplerosis. This leads to morphologically abnormal mitochondria with disorganized cristae, decreased production of ATP (red arrows) and exaggerated generation of ROS/oxidative stress, ultimately causing severe organ dysfunctions that primarily affect brain, liver, and kidney.

**Figure 2 cells-11-03179-f002:**
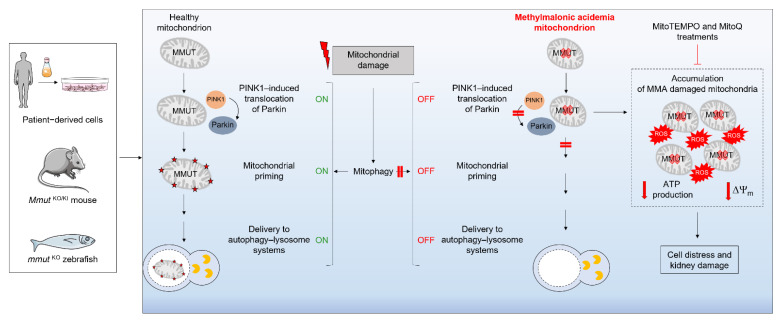
Working model depicting the link between MMUT, mitochondria, mitophagy, and epithelial homeostasis in healthy and MMA-affected kidney cells. In MMA-affected kidney cells and zebrafish, the deficiency of the enzyme MMUT and the resulting accumulation of toxic organic acids trigger mitochondrial abnormalities, which are characterized by a collapse of the mitochondrial membrane potential (ΔΨm, red arrows), impaired ATP production/bioenergetics (red arrows) and augmented generation of mitochondrial ROS and oxidative stress. Faulty execution of PINK1‒Parkin-mediated mitophagy induced by MMUT deficiency impedes the delivery of damaged mitochondria and their dismantling by autophagy–lysosome degradation systems. This, in turn, promotes the accumulation of damaged and/or dysfunctional, ROS-overproducing mitochondria that, ultimately, trigger cellular and kidney damage. The treatments with mitochondria-targeted ROS scavengers mito-TEMPO or MitoQ repair mitochondrial dysfunctions, neutralizes epithelial damage in MMA cells, and improves disease-relevant phenotypes in *mmut*-deficient zebrafish.

## Data Availability

Not applicable.

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
