# Peer review of "Mitochondrial Distress in Methylmalonic Acidemia: Novel Pathogenic Insights and Therapeutic Perspectives"

_cells, 2022, doi:10.3390/cells11193179_

Round 1

Reviewer 1 Report

The authors highlighted the biological functions of mitochondria-localized enzymes and discussed how deficit in these enzymes could have a pathogenic role in different inherited disorders.  Moreover, using methylmalonic acidemia (MMA) as a paradigm of mitochondrial dysfunction, they discussed how mitochondrial directed-signaling pathways sustain the physiological homeostasis of specialized cell types and how these may be disturbed in disease conditions. The authors also suggested how defects in autophagy-mediated quality control and surveillance systems could contribute to cellular dysfunction and indicated potential therapeutic strategies for affected tissues.

To study in deep the involvement of collateral pathways could be useful for the development of new therapeutics and this review will give a good contribution to this purpose.

I only suggest very few editing corrections:

Line 68: “The mitochondria are dynamic and plastic organelles harbour multiple copies..” this sentence has to be changed to “The mitochondria are dynamic and plastic organelles harboring multiple copies..”

 Line 119: the sentence “The priming of damaged and/or dysfunctional for autophagic degradation requires…” has to be changed to “The priming of damaged and/or dysfunctional mitochondria for autophagic degradation requires…”

Lines 132, 144: a square bracket is missing

Author Response

Itemized point-by-point response

We would like to express our gratitude for the time and effort of each reviewer and their constructive criticism and feedback. We have addressed their comments providing additional evidence and studies that would increasingly strengthen the message of our work and its readability.

We hope that the revised manuscript will be considered for publication.

Below each reviewer comment is our response written in blue.

New additions in the main manuscript text have been written in red.

Reviewer 1

The authors highlighted the biological functions of mitochondria-localized enzymes and discussed how deficit in these enzymes could have a pathogenic role in different inherited disorders.  Moreover, using methylmalonic acidemia (MMA) as a paradigm of mitochondrial dysfunction, they discussed how mitochondrial directed-signaling pathways sustain the physiological homeostasis of specialized cell types and how these may be disturbed in disease conditions. The authors also suggested how defects in autophagy-mediated quality control and surveillance systems could contribute to cellular dysfunction and indicated potential therapeutic strategies for affected tissues.

To study in deep the involvement of collateral pathways could be useful for the development of new therapeutics and this review will give a good contribution to this purpose.

*** We sincerely thank the Reviewer and are very pleased to hear that our manuscript would benefit the field.

I only suggest very few editing corrections:

Line 68: “The mitochondria are dynamic and plastic organelles harbour multiple copies..” this sentence has to be changed to “The mitochondria are dynamic and plastic organelles harboring multiple copies..”

Line 119: the sentence “The priming of damaged and/or dysfunctional for autophagic degradation requires…” has to be changed to “The priming of damaged and/or dysfunctional mitochondria for autophagic degradation requires…”

*** Thank you for pointing this out. As suggested, we have adjusted the wordings in the revised manuscript.

Lines 132, 144: a square bracket is missing

*** Thank you. Done.

Reviewer 2 Report

In the manuscript: “Mitochondrial distress in methylmalonic acidemia: Novel pathogenic insights and therapeutic perspectives,” the authors discuss 

 methylmalonic acidemia (MMA) as mitochondrial dysfunction and defects in autophagy-mediated quality cell surveillance. An interesting topic that contributes to knowledge in the area, but certain issues must be corrected.

1.    References must be reviewed since, in some sections of the manuscript, there are no references to which the reader can go for more information. For example, the paragraph in lines 209-224 contains no references.

2.    Why are kidney tubular cells important in MMA? It seems that this part needs to be defined and expanded so that the reader can clearly understand why the authors emphasize these cells during MMA.

3.    The authors must mention how MMUT deficiency is associated with ATP deficiencies and the production of ROS since neither in Figure 1 nor in the text is it mentioned how the deficiency of this enzyme induces these essential processes for cellular homeostasis.

4.    The authors should emphasize in the introduction the complications that MMA patients have. This part is briefly mentioned in section 5; however, in the introduction, it is highly recommended to describe the MMA disease and its ailments. Furthermore, the authors must mention why the kidney and liver are the most affected targets in MMA.

5.    Authors must mention why ROS produced due to MMA might be associated with mitochondrial dysfunction in a cycle where ROS induce mitochondrial dysfunction, and mitochondrial dysfunction induces ROS. It is important to mention this part since section 5, which talks about therapies associated with MMA, is very much related to the reduction of ROS.

6.    The authors must mention that probable mechanisms are involved in the deficiency of autophagic processes in MMA, of course, supported by the available bibliography.

Author Response

Itemized point-by-point response

We would like to express our gratitude for the time and effort of each reviewer and their constructive criticism and feedback. We have addressed their comments providing additional evidence and studies that would increasingly strengthen the message of our work and its readability.

We hope that the revised manuscript will be considered for publication.

Below each reviewer comment is our response written in blue.

New additions in the main manuscript text have been written in red.

In the manuscript: “Mitochondrial distress in methylmalonic acidemia: Novel pathogenic insights and therapeutic perspectives,” the authors discuss methylmalonic acidemia (MMA) as mitochondrial dysfunction and defects in autophagy-mediated quality cell surveillance. An interesting topic that contributes to knowledge in the area, but certain issues must be corrected.

*** We sincerely thank the Reviewer and are very pleased to hear that our manuscript would benefit the field.

References must be reviewed since, in some sections of the manuscript, there are no references to which the reader can go for more information. For example, the paragraph in lines 209-224 contains no references.

*** Thank you for pointing this out. We have adequately acknowledged and cited our and other studies.

Please see lines 224-239, in the revised manuscript.

Why are kidney tubular cells important in MMA? It seems that this part needs to be defined and expanded so that the reader can clearly understand why the authors emphasize these cells during MMA.

*** Thank you for pointing this out. We have now explained the role of mitochondria in the maintenance of energy and its implications in kidney tubular cell physiology and disease.

Please see lines 58-61, in the revised manuscript.

The authors must mention how MMUT deficiency is associated with ATP deficiencies and the production of ROS since neither in Figure 1 nor in the text is it mentioned how the deficiency of this enzyme induces these essential processes for cellular homeostasis.

*** Thank you for pointing this out. As suggested, we have added this information in the main text and respective figure.

Please see the lines 190-194 and 225-226, in the revised manuscript.

The authors should emphasize in the introduction the complications that MMA patients have. This part is briefly mentioned in section 5; however, in the introduction, it is highly recommended to describe the MMA disease and its ailments.

Furthermore, the authors must mention why the kidney and liver are the most affected targets in MMA.

*** Thank you for pointing this out. As suggested, we have included this background in the main text and figure.

Please see lines 61-69, in the revised manuscript.

Authors must mention why ROS produced due to MMA might be associated with mitochondrial dysfunction in a cycle where ROS induce mitochondrial dysfunction, and mitochondrial dysfunction induces ROS. It is important to mention this part since section 5, which talks about therapies associated with MMA, is very much related to the reduction of ROS.

*** Yes, we agree. We have now clarified this feed-forward loop occurring between MMUT enzyme, mitochondria, and generation of ROS/oxidative stress.

Please see the lines 190-194 in the revised manuscript.

The authors must mention that probable mechanisms are involved in the deficiency of autophagic processes in MMA, of course, supported by the available bibliography.

*** Thank you for pointing this out. The molecular underpinnings by which MMUT deficiency suppresses PINK1/Parkin-directed “eat me” signals and hence mitophagy, remain equally elusive. An intriguing scenario might be that MMUT deficiency might alter the stability of PINK1 by disabling the interaction with yet-to-be defined factors that protect PINK1 from processing and degradation. Alternatively, MMUT deficiency might trigger stress-related posttranslational modifications, such as S-nitrosylation or aberrant methylmalonylation that inhibits PINK1 kinase activity and hence mitophagy-directed degradation pathway.

We have now emphasized these possible mechanistic angles in the Conclusion remarks section.

Please see the lines, 365-371 in the revised manuscript.

Round 2

Reviewer 2 Report

Los autores realizaron las revisiones pedidas por lo que creo que el manuscrito puede ser publicado.

Author Response

We sincerely thank this reviewer and are delighted to hear that our work is suitable for being published.